

# Establishment of a 12-gene expression signature to predict colon cancer prognosis

Dalong Sun[1,*], Jing Chen[2,*], Longzi Liu[3,*], Guangxi Zhao[4], Pingping Dong[1], Bingrui Wu[5], Jun Wang[6] and Ling Dong[1]

[1] Department of Gastroenterology and Hepatology, Zhongshan Hospital, Fudan University, Shanghai, China
[2] Department of Neurology, Shanghai Fifth People's Hospital, Fudan University, Shanghai, China
[3] Department of Hepatic Surgery, Liver Cancer Institute, and Key Laboratory of Carcinogenesis and Cancer Invasion (Ministry of Education), Zhongshan Hospital, Fudan University, Shanghai, China
[4] Department of Gastroenterology, Shanghai East Hospital, Tongji University School of Medicine, Shanghai, China
[5] Key Laboratory of Glycoconjugate Research Ministry of Public Health, Department of Biochemistry and Molecular Biology, Shanghai Medical College, Fudan University, Shanghai, China
[6] Guangzhou Institute of Pediatrics, Guangzhou Women and Children's Medical Center, Guangzhou Medical University, Guangzhou, Guangdong Province, China
[*] These authors contributed equally to this work.

Corresponding authors
Ling Dong,
dong.ling@zs-hospital.sh.cn
Jun Wang, jwang03@sibs.ac.cn

## ABSTRACT

A robust and accurate gene expression signature is essential to assist oncologists to determine which subset of patients at similar Tumor-Lymph Node-Metastasis (TNM) stage has high recurrence risk and could benefit from adjuvant therapies. Here we applied a two-step supervised machine-learning method and established a 12-gene expression signature to precisely predict colon adenocarcinoma (COAD) prognosis by using COAD RNA-seq transcriptome data from The Cancer Genome Atlas (TCGA). The predictive performance of the 12-gene signature was validated with two independent gene expression microarray datasets: GSE39582 includes 566 COAD cases for the development of six molecular subtypes with distinct clinical, molecular and survival characteristics; GSE17538 is a dataset containing 232 colon cancer patients for the generation of a metastasis gene expression profile to predict recurrence and death in COAD patients. The signature could effectively separate the poor prognosis patients from good prognosis group (disease specific survival (DSS): Kaplan Meier (KM) Log Rank $p = 0.0034$; overall survival (OS): KM Log Rank $p = 0.0336$) in GSE17538. For patients with proficient mismatch repair system (pMMR) in GSE39582, the signature could also effectively distinguish high risk group from low risk group (OS: KM Log Rank $p = 0.005$; Relapse free survival (RFS): KM Log Rank $p = 0.022$). Interestingly, advanced stage patients were significantly enriched in high 12-gene score group (Fisher's exact test $p = 0.0003$). After stage stratification, the signature could still distinguish poor prognosis patients in GSE17538 from good prognosis within stage II (Log Rank $p = 0.01$) and stage II & III (Log Rank $p = 0.017$) in the outcome of DFS. Within stage III or II/III pMMR patients treated with Adjuvant Chemotherapies (ACT) and patients with higher 12-gene score showed poorer prognosis (III, OS: KM Log Rank $p = 0.046$; III & II, OS: KM Log Rank $p = 0.041$). Among stage II/III pMMR patients with lower 12-gene scores in GSE39582, the subgroup receiving ACT showed significantly longer OS time compared with those who received no ACT (Log Rank $p$
$= 0.021$), while there is no obvious difference between counterparts among patients with higher 12-gene scores (Log Rank $p = 0.12$). Besides COAD, our 12-gene signature is multifunctional in several other cancer types including kidney cancer, lung cancer, uveal and skin melanoma, brain cancer, and pancreatic cancer. Functional classification showed that seven of the twelve genes are involved in immune system function and regulation, so our 12-gene signature could potentially be used to guide decisions about adjuvant therapy for patients with stage II/III and pMMR COAD.

## INTRODUCTION

Colorectal cancer (CRC) is one of the most common cancers in men and women, representing almost 10% of the global cancer incidents and the third leading cause of cancer death worldwide (*McGuire, 2016*). CRC comprises three different subtypes according to distinct pathway operate: chromosomal-instable, microsatellite-instable, and CpG island methylator phenotype, all of which differ in morphology, genetic background, molecular profile, clinical behavior, and response to therapy (*De Sousa et al., 2013*). Current prognostic model based on the classic tumor-node-metastasis (TNM) staging is the standard prognosis factor for CRC in clinical practice. However, due to the high heterogeneity of disease, the patients at similar stage behave differently in terms of recurrence and response to chemotherapy often differs. Better parameters to guide patients' prognostic stratification and personalized medicine are urgently needed. Currently, some prognostic and predictive molecular markers have been developed. Microsatellite instability (MSI) is the molecular hallmark of DNA mismatch repair (MMR) deficiency. In stage II of the disease, MSI status helps select patients with high risk of developing recurrence (*Brychtova et al., 2017*). MSI status can also be a predictor of the benefit of adjuvant chemotherapy with fluorouracil in stage II and stage III colon cancer (*Ribic et al., 2003*). KRAS mutation status has been validated as a molecular marker for prediction of non-response to EGFR targeted drugs in metastatic CRC (*Cunningham et al., 2010*; *Karapetis et al., 2008*; *Siena et al., 2009*). However, due to complex pathways contributing to cancer progression, single molecular marker might not be efficient enough to predict prognosis and individualize in selecting adjuvant therapy.

The development of gene expression profiling technologies such as microarray and Next Generation Sequencing (NGS) provide further opportunities to comprehensively characterize the molecular features of cancer. Gene-expression profiling has been used to develop genomic tests that may provide better predictions of clinical outcomes in combination with traditional clinicopathologic factors (*Gray et al., 2007*; *Venook et al., 2011*; *Meropol et al., 2011*; *Ebata, Hirata & Kawauchi, 2016*; *Guinney et al., 2015*; *Marisa et al., 2013*; *Smith et al., 2010*; *Gentles et al., 2015*). Some commercially genomic assays are

available for the prediction of clinical outcome in CRC patients. The most well-known one is the Oncotype DX Colon Cancer Assay, which is a 12-gene (seven cancer related genes and five reference genes) genomic test that has been used to help identify individuals with high recurrence risk from stage II colon cancer patients with T3 and MMR proficient tumors (*Gray et al., 2007*; *Venook et al., 2011*; *Meropol et al., 2011*). However, the five reference genes in Oncotype DX Assay contain PGK1 and GPX1, which are important players in the process of energy metabolism and cellular oxidative stress, both of which are actively involved in cancer development and metastasis (*Ebata, Hirata & Kawauchi, 2016*; *Moloney & Cotter, 2017*). Normalization with PGK1 and GPX1 might have diluted the tumorous heterogeneities among cancer patients. In this work, we applied two steps of supervised machine-learning method and established a 12-gene expression signature to precisely predict colon adenocarcinoma (COAD) prognosis by exhaustively using expression of all genes of The Cancer Genome Atlas (TCGA) COAD patients.

## MATERIALS AND METHODS

### TCGA and GEO datasets

RNA-seq data and clinic information for all cancer types were obtained from TCGA RNA-seq database (https://cancergenome.nih.gov/). Microarray expression data and clinic information for COAD patients were retrieved from Gene Expression Omnibus (GEO) database (https://www.ncbi.nlm.nih.gov/geo/).

### Development of the gene expression signature

The development process has a training and validation phase.

### Training stage has two phases
#### *Phase I*
#### *Grouping*

The TCGA COAD patients were used for the development of prototype of the 118-gene signature that could predict COAD prognosis. We applied a similar supervised machine learning method that was used for MammaPrint (*Van et al., 2002*). Forty-two patients that experienced relapse within three years were designated as poor prognosis. Forty-nine patients who were relapse free for at least three years were categorized as good prognosis. The gene expression values were centered and scaled before grouping. For the training dataset, 32 and 39 patients were randomly chosen from poor and good prognosis category, respectively. The rest of the patients were grouped as test dataset. Detailed clinic information is listed in Table S1.

#### *Selection of genes with high correlation to real prognosis status*

Overall, there are 20,530 genes in the raw RNA-seq data. The Pearson correlation coefficients with real prognosis status were calculated for all genes. Genes with absolute correlation coefficient greater than 0.3 were selected. To test whether such correlation coefficient distribution could be found by chance, a permutation method was used to generate 10,000 Monte-Carlo simulations randomizing the correlation between gene expression data of the 71 training patients and corresponding prognostic categories.

### Supervised machine-learning method

Gene number incorporated in the signature needs to be optimized. One thousand, five hundred and ten genes were reordered by absolute coefficients from maximum to minimum. Starting from the top two genes on the list, 755 signatures were generated by adding two more genes from the top list each time until all the 1,510 genes were exhaustively used as reporters. A Leave-One-Out Cross-Validation (LOOCV) method was employed to check the performances of these signatures:

Step 1: leave one tumor out;

Step 2: calculate the good- and poor-prognosis expression template by averaging the expression values for each gene incorporated in good-prognosis group and poor-prognosis group, respectively. Then we defined a parameter called risk coefficient (risk-coef.). For a tumor, risk coefficient was calculated with its gene expression profile and good- and poor-prognosis expression template:

Risk-coef = cor-coef. to good template − cor-coef. to poor template;

Step 3: calculate the risk-coefs for all the remaining 70 training samples and the left out sample. Reorder the 71 samples by ranking their risk-coefs from small to large. Determine the genomic risk by taking first 32 tumors as high genomic risk and the rest 39 as low genomic risk. Check the consistency between genomic risk and real risk for the left out sample;

Step 4: repeat step 1–3 iteratively until all the 71 samples have been left out once. Collect the error counts when there is a disagreement between genomic risk and real risk for the left out sample.

Better signatures with least error counts were selected.

### Cross-validation without information leak

The 1,510 genes were obtained using all training samples including the one left out for cross validation, so there might be an over-fitting issue due to information leak. A modified LOOCV with no information leak was performed as below:

Step 1: leave one patient out;

Step 2: calculate the Pearson correlation coefficients with real prognosis status for all genes with the reminding 70 training samples. Filter the genes with |coefficient| $\geq 0.3$.

Step 3: generate the signature with the genes selected and predict the genomic risk for the left out sample.

Step 4: repeat step 1–3 iteratively until all the 71 samples have been left out once.

### Phase II

Further machine learning process was applied to generate a concise scoring system. Before machine learning, the RPKM (Reads Per Kilobase per Million mapped reads) values need normalization, which was done through dividing them by geometric mean of RPKM values of TFRC, GUSB, and RPLP0. Firstly, the TCGA COAD patients (Table S2) were split into training and test dataset. There is no significant difference between the clinicopathologic factors of these two groups (Table 1). For each of the 118 genes, we calculated the coefficient and $p$-value in univariate Cox Proportional Hazard regression model (CPH) with training dataset. Then we reordered the gene list by sorting the univaraite Cox-regression $p$-value

**Table 1  Clinicopathologic features of 240 TCGA COAD patients.**

| Characteristic | Training set ($N = 119$) No. of patients (%) | Testing set ($N = 121$) No. of patients (%) | p value |
|---|---|---|---|
| Age (mean $\pm$ SD) | 66.4 $\pm$ 13.0 | 63.2 $\pm$ 13.8 | 0.069[a] |
| **Gender** | | | |
| Male | 60 (50.4%) | 54 (44.6%) | 0.438[b] |
| Female | 59 (49.6%) | 67 (55.4%) | |
| **Stage** | | | |
| I | 20 (16.8%) | 20 (16.5%) | |
| II | 47 (39.5%) | 48 (39.7%) | 0.998[c] |
| III and IV | 52 (43.7%) | 53 (43.8%) | |
| **Primary tumor** | | | |
| T1 and T2 | 20 (16.8%) | 23 (19.0%) | 0.737[b] |
| T3 and T4 | 99 (83.2%) | 98 (81.0%) | |
| **Microsatellite status** | | | |
| MSI-L | 23 (19.3%) | 23 (19.0%) | |
| MSI-H | 20 (16.8%) | 20 (16.5%) | 0.995[c] |
| MSS | 76 (63.9%) | 78 (64.4%) | |
| **Lymphatic_invasion** | | | |
| No | 77 (64.7%) | 77 (63.6%) | 0.999[b] |
| Yes | 33 (27.7%) | 34 (28.1%) | |
| *Unknown* | *9 (7.6%)* | *10 (8.3%)* | Excluded |

**Notes.**
[a] *t* test.
[b] Fisher's exact test.
[c] Chi-squared test.

from minimum to maximum. So the top genes have stronger correlations with cancer prognosis. Starting from the top one gene in the list, we added one more gene iteratively from the top for multivariate CPH analysis. In every iteration step, the fitness of established signature on test dataset was checked by calculating Kaplan Meier Log Rank p-value (KM-p). At the end of iteration, signature incorporating the top 12 genes has the minimum test dataset KM-p and was deemed as the optimal one. The multivariate Cox coefficient of each gene in the final signature was extracted to generate the scoring system:

$$\text{Riskscore} = \sum_{i=1}^{n} Ei * \beta i.$$

*Ei*: expression level of gene *i*; $\beta i$: multivariate Cox-regression coefficient of gene *i*.

## Validation stage

The GEO microarray datasets were used to validate the final gene expression signature. For genes with more than one probe, the probe showing minimum univariate CPH p-value was selected. Relative expression level was obtained via dividing the probe signal by geometric mean of signals of TFRC, GUSB, and RPLP0. For each tumor, a risk score was obtained by calculating the weighted summation of relative expressions of the 12-gene. For a certain dataset, patients with risk scores below the median value of the population were designated

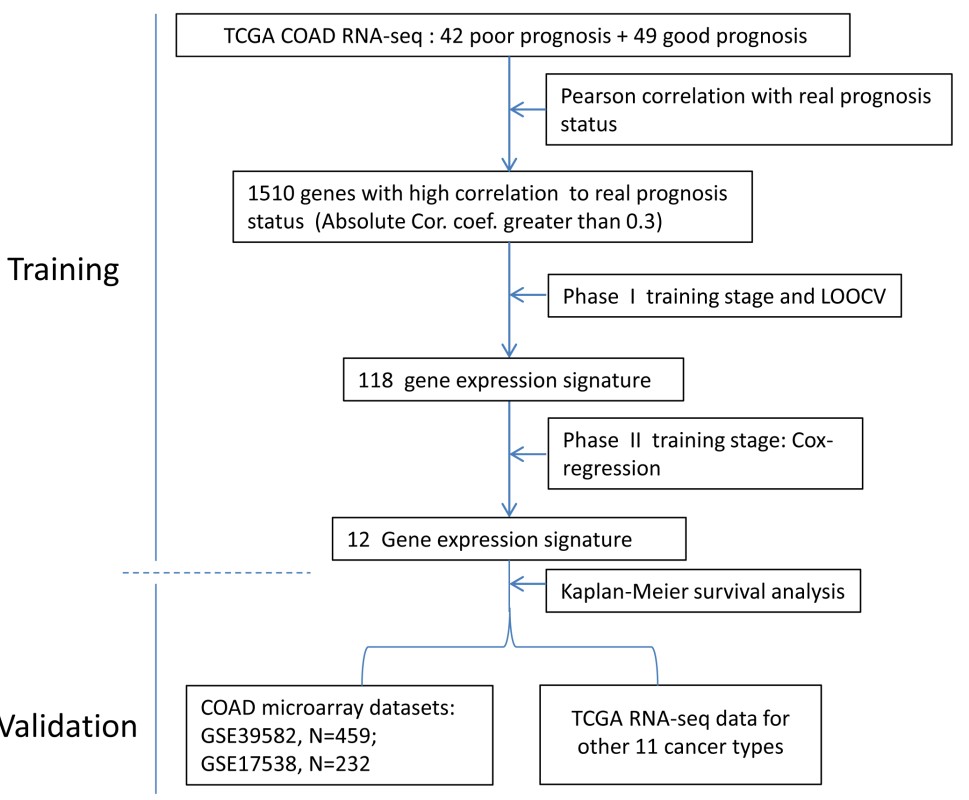

**Figure 1  The flow chart of the development process of the COAD gene expression signature.**

as the low risk group, while the rest of the patients were categorized as the high risk group. Survival comparisons between high and low risk groups were conducted by Kaplan–Meier plotting. Log Rank $p$ value <0.05 was considered as significantly different. Other cancer types in TCGA library were also retrieved to validate the 12-gene signature.

# RESULTS

## Development of signature prototype

The development process was shown as the flow chart in Fig. 1. With the TCGA COAD data, an unbiased screening method was used to obtain 1,510 genes showing absolute correlations greater than 0.3 with disease outcomes. The frequency distribution of number of genes with absolute coefficient no less than 0.3 in the 10,000 Monte-Carlo trials was displayed in Fig. S1. The probability of obtaining 1,510 genes or more with an absolute correlation coefficients of at least 0.3 with prognosis categories purely by chance was 0.0019 ($p < 0.05$), which was fair for us to reject the null hypothesis.

During the (Leave-One-Out Cross Validation) LOOCV process, 755 signatures were generated. Least violation times were observed when signature employed the top 16, 36, 40, 42, 44, 46, 48, 50, 56, 58, 60, 62, 64, 66, 68, 70, 72, 74, 76, 78, 80, 82, 84, 86, or 118 genes. We further found that the predictive accuracy rates were high towards the 71 training samples with the signature containing the top 118 genes (Fig. 2). We had the luxury to
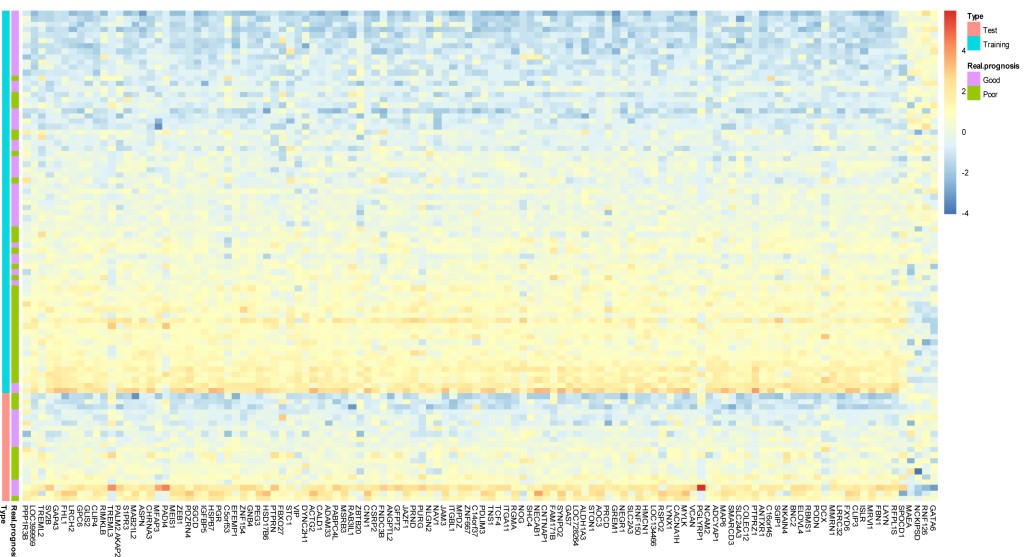

**Figure 2 Prototype of the gene expression signature.** Expression heatmap plotting of 118 prognostic marker genes in training dataset and 20 patients in test dataset. Each row represents an observation (patient) and each column is a gene, whose name is labeled at the bottom. Tumors are ordered by the correlation to the average expression pattern of the good and poor prognosis group. Genes are ordered by their correlation coefficients with the two prognosis categories. The real prognosis status for each tumor is displayed in the middle panel.

further validate the established signatures using the remaining 20 independent samples in test dataset. For each signature, receiver operating characteristic curve (ROC) was plotted with the information of risk-coefs and real risk of the 91 TCGA patients to compare the performances of the 25 signatures. There was no significant difference among the performances of these signatures (Fig. 3 and Table 2).

Because the above 1,510 genes were obtained using all the training samples including the one left out for cross validation, a modified LOOCV without information leak was performed. Seventy-one additional signatures were created. The vast majority of the original 1,510 genes were shared by most of the 71 signatures (Fig. S2). So there was very limited information leak introduced during the previous training process.

## Development of 12-gene signature

For the purpose of concise and simplicity, we further established a 12-gene expression signature based on the 118 genes obtained in phase I training stage. Expressional coefficients were assigned to respective genes. Each patient has a risk score by calculating the weighted summation of expression values of the 12 genes. The Kaplan–Meier (KM) survival analysis showed that among TCGA COAD patients, the high risk group displayed significantly poorer prognosis than low risk group regarding to disease free survival (DFS) (training dataset: KM Log Rank $p = 0.0001$; test dataset: KM Log Rank $p = 0.0005$) (Fig. 4).

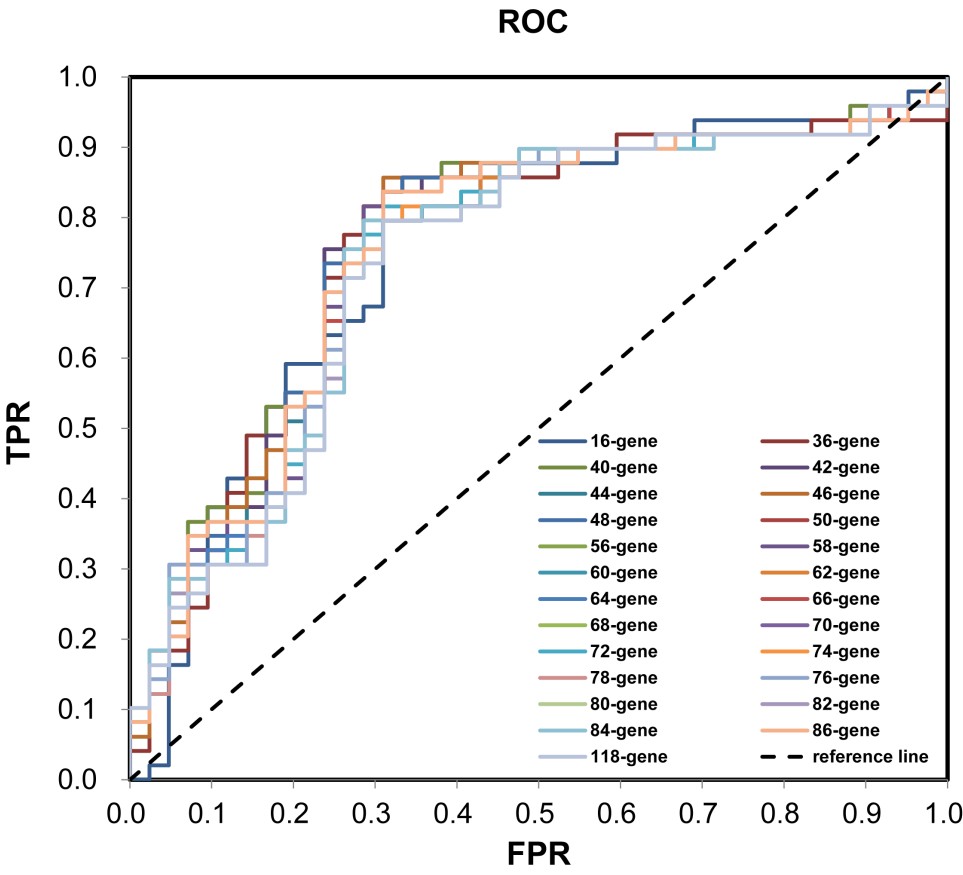

**Figure 3    ROC plotting for the 25 signatures generated during phase I training process.** ROC with the information of risk-coefs and real risk of the 91 TCGA patients. ROC, receiver operating characteristic curve. TPR, true positive rate. FPR, false positive rate.

## Prognostic values of the 12-gene signature in other COAD datasets

GSE17538 (GSE17536 and GSE17537) was used to validate the 12-gene expression signature. With both clinic information and microarray gene expression of 232 colon cancer patients, *Smith et al. (2010)* established a metastasis gene expression profile to predict recurrence and death in COAD patients. The 12-gene signature could effectively separate the poor prognosis patients from good prognosis group (Figs. 5A–5C, Disease specific survival (DSS): KM Log Rank $p = 0.0034$; Overall survival (OS): KM Log Rank $p = 0.0336$; Disease free survival (DFS): KM Log Rank $p = 0.0004$). After stage stratification, the signature could still distinguish poor prognosis patients from good within stage II (Fig. 5D, Log Rank $p = 0.01$) and stage II & III (Fig. 5E: Log Rank $p = 0.017$) in terms of DFS.

GSE39582 is a dataset including 566 COAD cases and 19 non-tumoral colorectal mucosas. With this dataset, Marisa et al. developed gene expression classification of colon cancer defining six molecular subtypes with distinct clinical, molecular and survival characteristics (*Marisa et al., 2013*). In patients with proficient mismatch repair system (pMMR), our 12-gene signature could effectively distinguish high risk group from low

**Table 2  Statistics of the ROC analysis.**

| Signature | AUC | SE | Progressive p | Progressive 95% CIs | |
|---|---|---|---|---|---|
| | | | | Lower bound | Upper bound |
| 16-gene | 0.7517 | 0.0531 | 0.0000 | 0.6476 | 0.8558 |
| 36-gene | 0.7600 | 0.0529 | 0.0000 | 0.6562 | 0.8637 |
| 40-gene | 0.7653 | 0.0520 | 0.0000 | 0.6634 | 0.8672 |
| 42-gene | 0.7609 | 0.0525 | 0.0000 | 0.6581 | 0.8638 |
| 44-gene | 0.7604 | 0.0525 | 0.0000 | 0.6576 | 0.8633 |
| 46-gene | 0.7614 | 0.0524 | 0.0000 | 0.6588 | 0.8641 |
| 48-gene | 0.7575 | 0.0528 | 0.0000 | 0.6540 | 0.8610 |
| 50-gene | 0.7541 | 0.0530 | 0.0000 | 0.6503 | 0.8580 |
| 56-gene | 0.7493 | 0.0532 | 0.0000 | 0.6450 | 0.8536 |
| 58-gene | 0.7488 | 0.0533 | 0.0000 | 0.6444 | 0.8532 |
| 60-gene | 0.7483 | 0.0532 | 0.0000 | 0.6439 | 0.8527 |
| 62-gene | 0.7478 | 0.0533 | 0.0000 | 0.6433 | 0.8524 |
| 64-gene | 0.7468 | 0.0534 | 0.0001 | 0.6421 | 0.8515 |
| 66-gene | 0.7459 | 0.0535 | 0.0001 | 0.6409 | 0.8508 |
| 68-gene | 0.7449 | 0.0534 | 0.0001 | 0.6402 | 0.8496 |
| 70-gene | 0.7459 | 0.0534 | 0.0001 | 0.6412 | 0.8505 |
| 72-gene | 0.7468 | 0.0534 | 0.0001 | 0.6422 | 0.8514 |
| 74-gene | 0.7444 | 0.0535 | 0.0001 | 0.6395 | 0.8493 |
| 76-gene | 0.7444 | 0.0535 | 0.0001 | 0.6396 | 0.8492 |
| 78-gene | 0.7430 | 0.0537 | 0.0001 | 0.6377 | 0.8482 |
| 80-gene | 0.7410 | 0.0539 | 0.0001 | 0.6354 | 0.8467 |
| 82-gene | 0.7420 | 0.0538 | 0.0001 | 0.6365 | 0.8474 |
| 84-gene | 0.7410 | 0.0539 | 0.0001 | 0.6353 | 0.8467 |
| 86-gene | 0.7410 | 0.0539 | 0.0001 | 0.6354 | 0.8466 |
| 118-gene | 0.7347 | 0.0542 | 0.0001 | 0.6284 | 0.8410 |

**Notes.**
AUC, Area-Under-Curve; SE, Standard Error; 95% CIs, 95% Confidence Intervals.

risk group (Figs. 6A and 6B, Relapse free survival (RFS): KM Log Rank $p = 0.022$; OS: KM Log Rank $p = 0.005$). No significant difference was found in KM analysis performed among dMMR patients. Further survival analysis was performed within stage III or II & III and pMMR patients treated with Adjuvant Chemotherapies (ACT): patients with higher 12-gene score showed poorer prognosis (Figs. 6C and 6D: III, OS: KM Log Rank $p = 0.046$; III & II, OS: KM Log Rank $p = 0.041$). Interestingly, among stage II & III pMMR patients with lower 12-gene scores, subgroup receiving ACT showed significantly longer OS time compared with those who received no ACT (Fig. 6E: Log Rank $p = 0.021$), while there is no obvious difference between counterparts among patients with higher 12-gene scores (Fig. 6F: Log Rank $p = 0.12$).

Interestingly, advanced stage patients were significantly enriched in high 12-gene score group (Table 3).

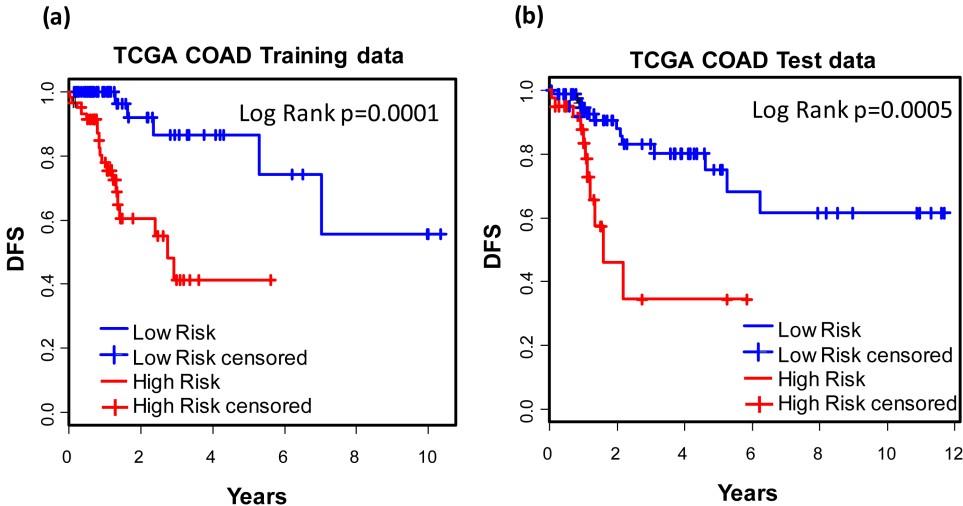

**Figure 4 Prognostic values of the 12-gene signature.** Kaplan–Meier analysis of the high and low 12-gene risk score patients among TCGA COAD patients in training (A) and test dataset (B) in phase II training stage.

## Predictive performances of the 12-gene signature in other cancer types

We also tested the performance of the signature in other cancer types. TCGA RNA-seq data and corresponding clinic information for 24 cancer types were retrieved for validation. Surprisingly, KM results showed that our signature successfully separated good prognosis patients from poor prognosis patients in several other cancer types including pan-kidney cohort (KIPAN) (Fig. 7A, OS: KM Log Rank $p = 6.815e − 6$), kidney renal clear cell carcinoma (KIRC) (Fig. 7B, DFS: KM Log Rank $p = 0.0480$), kidney renal papillary cell carcinoma (KIRP) (Fig. 7C, DFS: KM Log Rank $p = 0.0027$; Fig. 7D OS: Log Rank $p = 0.0129$), lung squamous cell carcinoma (LUSC) (Fig. 7E, DFS: Log Rank $p = 0.0071$), and skin cutaneous melanoma (SKCM) (Fig. 7F, DFS: Log Rank $p = 0.01117$), brain lower grade glioma (LGG) (Fig. 8A, OS: Log Rank $p = 0.0031$), uveal melanoma (UVM) (Fig. 8B, OS: Log Rank $p = 0.0054$), glioblastoma (GBM) (Fig. 8C, OS: Log Rank $p = 0.0074$), cervical and endocervical cancers (CESC) (Fig. 8D, OS: Log Rank $p = 0.0090$), pancreatic adenocarcinoma (PAAD) (Fig. 8E, OS: Log Rank $p = 0.0127$), stomach adenocarcinoma (STAD) (Fig. 8F, OS: Log Rank $p = 0.0456$).

## DISCUSSION

Numerous attempts have been made to establish gene expression signatures for the purpose of precise prediction of colorectal cancer prognosis (*Gray et al., 2007*; *Venook et al., 2011*; *Meropol et al., 2011*; *Ebata, Hirata & Kawauchi, 2016*; *Guinney et al., 2015*; *Marisa et al., 2013*; *Smith et al., 2010*; *Gentles et al., 2015*). A meta-analysis was done to assess the clinical value of several published prognosis gene expression signatures in colorectal cancer (*Sanz-Pamplona et al., 2012*). Although most of the published signatures showed significant

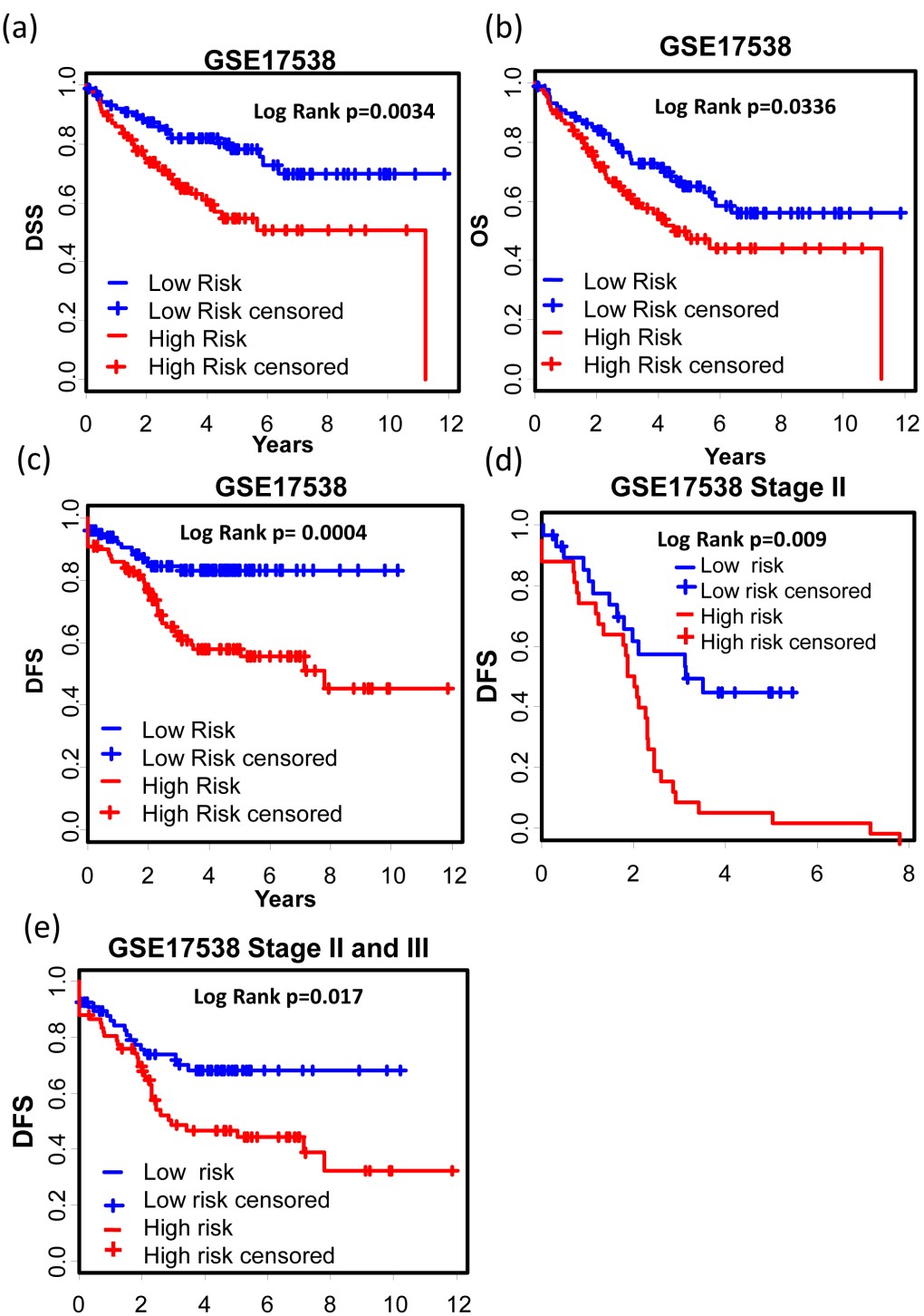

**Figure 5   Prognostic values of the 12-gene signature in other COAD datasets.** (A)–(C) Kaplan–Meier curves showing patients (stage I–IV) with high and low 12-gene risk score in endpoints of DSS, OS, and DFS, respectively; Kaplan–Meier curves showing patients at stage II (D) or II & III (E) with high and low 12-gene risk score in terms of DFS. DFS, disease free survival; DSS, disease specific survival; OS, overall survival.

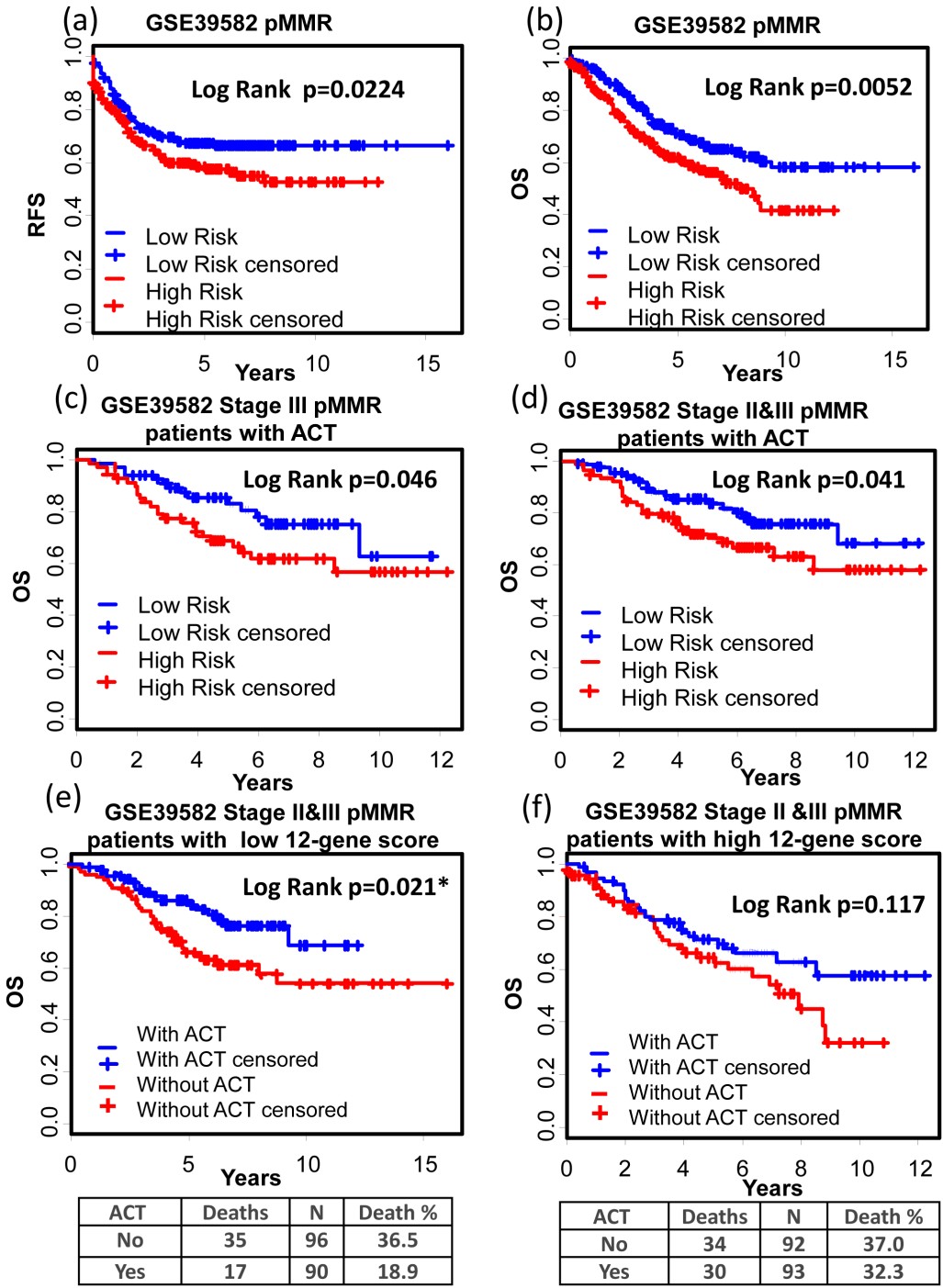

| ACT | Deaths | N | Death % |
|-----|--------|-----|---------|
| No | 35 | 96 | 36.5 |
| Yes | 17 | 90 | 18.9 |

| ACT | Deaths | N | Death % |
|-----|--------|-----|---------|
| No | 34 | 92 | 37.0 |
| Yes | 30 | 93 | 32.3 |

**Figure 6 Prognostic values of the 12-gene signature in GSE39582.** (A) and (B) Kaplan–Meier curves showing patients (stage I–IV) with high and low 12-gene risk score in endpoints of RFS and OS, respectively; Kaplan–Meier curves showing stage III (C) or II & III (D) pMMR patients (treated with ACT) with high and low 12-gene risk score in respect to the endpoint of OS; (E) in stage II & III pMMR patients with low 12-gene scores, ACT subgroup displayed better OS outcome than control; (F) in stage II & III pMMR patients with high 12-gene scores, ACT and control group displayed no significant difference in the outcome of OS. RFS: relapse-free survival; OS: overall survival; pMMR: proficient mismatch repair system.

**Table 3 Distribution of advanced stage patients between high- and low-score group.** Fisher's exact test was used for statistical analysis.

| Dataset | Group | Stage I & II | Stage III & IV | *p* value |
|---------|-------|--------------|----------------|-----------|
| GSE17538 | High score group | 19 (20%) | 78 (80%) | 0.0003 |
| | Low score group | 43 (44%) | 54 (56%) | |
| TCGA | High score group | 53 (49%) | 55 (51%) | 0.0277 |
| | Low score group | 69 (64%) | 38 (36%) | |

statistical association with prognosis, their accuracy to classify independent tumor samples into high-risk and low-risk group is limited. So we need more robust and accurate gene expression signature that can predict prognosis cross independent COAD datasets. Here we established a gene expression signature by applying two steps of supervised machine-learning method. The predicative accuracy of our gene expression signature was proven by validation in two large independent gene expression microarray datasets (GSE39582, $N = 459$; GSE17538, $N = 232$). Decision making regarding adjuvant therapy has been a debate among professional clinical organizations over the past 20 years (*Dotan & Cohen, 2011*; *Meropol, 2011*; *Vachani, 2013*). Currently speaking, uncertainty is present in adjuvant chemotherapeutic effects among stage II COAD patients who are mismatch repair system proficient. The Scottish Intercollegiate Guidelines Network (SIGN), ASCO, and NCCN are following different guidelines regarding this issue (*Gao et al., 2016*). Resectable COAD patients with pMMR routinely receive 5-FU based postoperative adjuvant chemotherapy (POCT) which has been shown to provide a relatively small absolute benefit (*Andre et al., 2009*; *Gill et al., 2004*; *Sargent et al., 2009*; *Gray et al., 2011*; *Alex et al., 2017*), indicating that many COAD patients might have been over-treated due to the lacking of an effective test to stratify the patients further. Our gene signature showed important prognostic value for stage II or/and III pMMR COAD patients. There validation results in GSE39582 indicate that lower 12-gene score patients have gained survival benefit from adjuvant chemotherapies, while high score patients treated with adjuvant chemotherapies didn't receive survival benefit. So our 12-gene signature could potentially be used to guide decisions about adjuvant therapy for patients with stage II & III and pMMR colon cancer.

Seven of the proteins encoded by the 12 genes were related to immune system, they are TREML2, PADI4, NCKIPSD, PTPRN, PGLYRP1, C5orf53, and TREML3, indicating the essential roles of deregulated immune response in COAD progression and metastasis (Table S3). TREML2, acting as the counter-receptor for B7-H3, promotes T cell responses (*Hashiguchi et al., 2008*). PADI4 protein catalyzes the conversion of arginine to citrulline residue. With specific high expression in blood lymphocytes (*Asaga et al., 2001*; *Anzilotti et al., 2010*), PADI4 is believed to be an active autoimmune player in synovial tissue of rheumatoid arthritis (*Chang et al., 2005*). It is reported that cell free circulation PADI4 mRNA level (together with cfDNA, PPBP, and haptoglobin) in peripheral blood of non-small cell lung cancer patients was significantly higher than that in healthy donors, so PADI4 may serve as a potential marker for NSCLC diagnosis (*Ulivi et al., 2013*). As a member of protein tyrosine phosphatase (PTP), PTPRN is an autoantigen in the sera of insulin-dependent diabetes mellitus (IDDM) patients, making it a promising therapeutic target

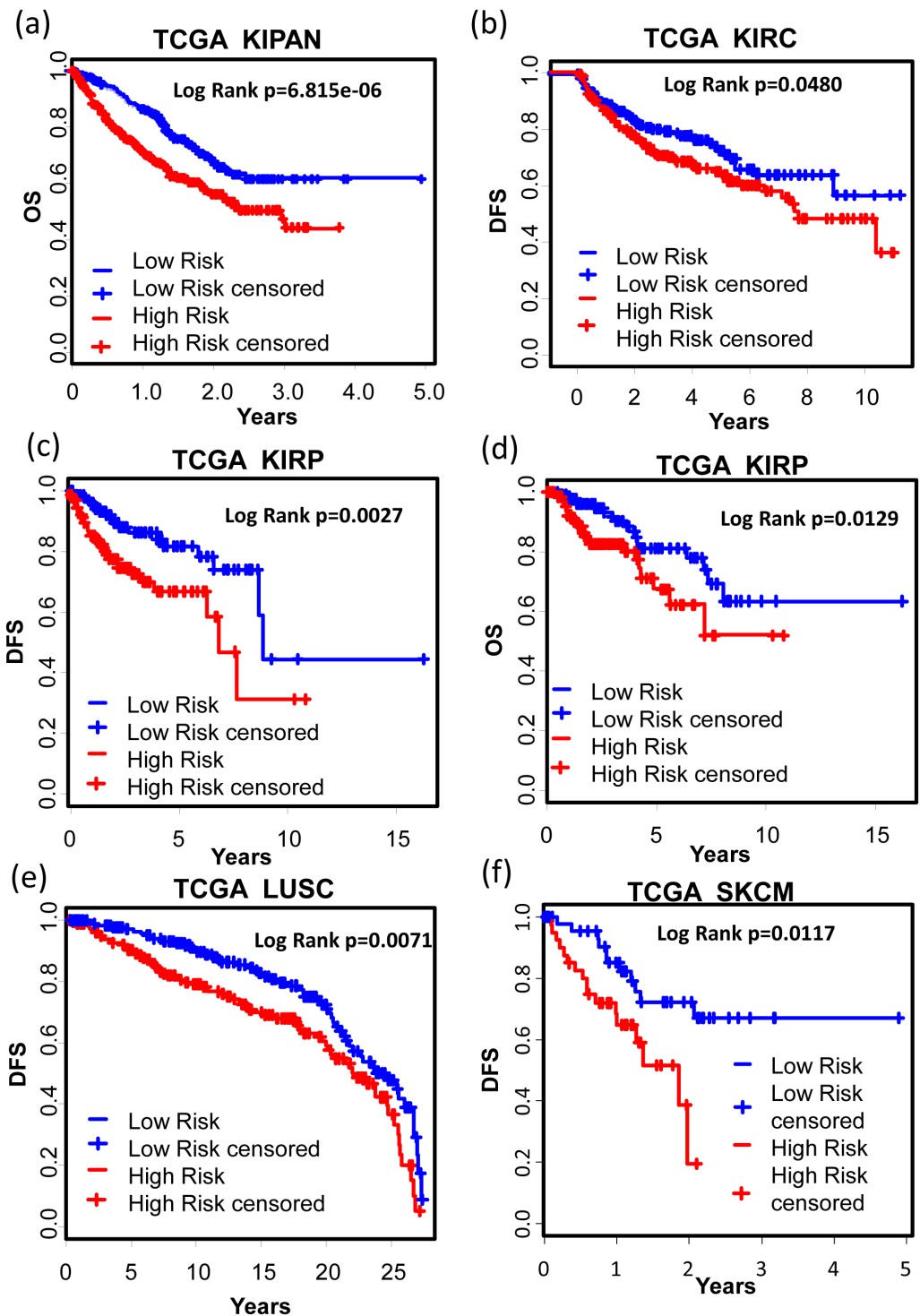

**Figure 7** **KM analysis of the high and low 12-gene risk score patients for the major outcomes in other cancer types.** (A) OS in pan-kidney cohort (KIPAN); (B) DFS in kidney renal clear cell carcinoma (KIRC). (C) & (D) DFS and OS in kidney renal papillary cell carcinoma (KIRP), respectively. (E) DFS in lung squamous cell carcinoma (LUSC). (F) DFS in skin cutaneous melanoma (SKCM). OS, overall survival. DFS, disease free survival.

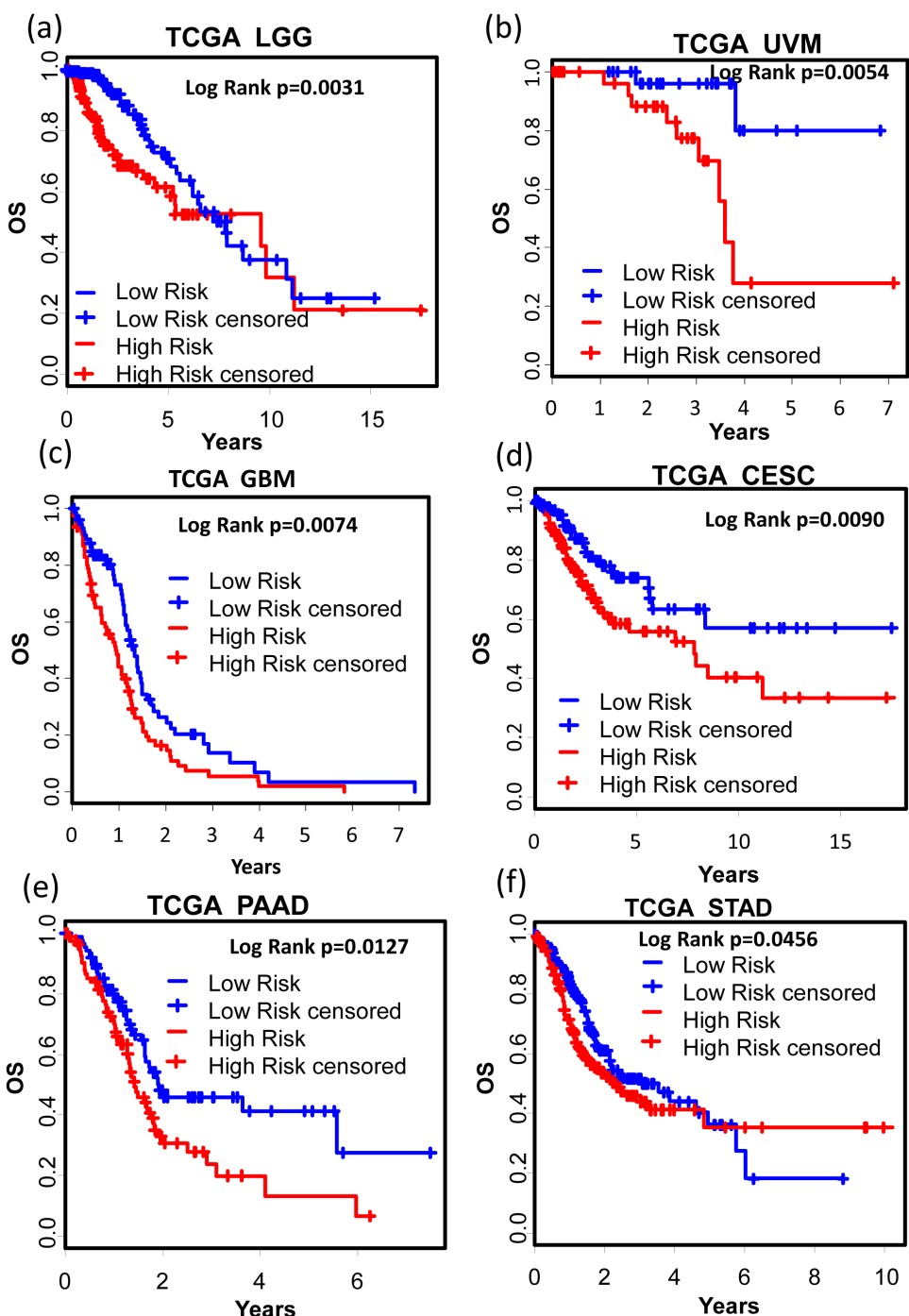

**Figure 8** Kaplan–Meier analysis of the high and low 12-gene risk score patients for the major outcomes in other cancer types. (A) OS in brain lower grade glioma (LGG). (B) OS in uveal melanoma (UVM). (C) OS in glioblastoma (GBM). (D) OS in cervical and endocervical cancers (CESC). (E) OS in pancreatic adenocarcinoma (PAAD). (F) OS in stomach adenocarcinoma (STAD). OS, overall survival.

of autoimmunity in IDDM (*Rabin et al., 1994*; *Solimena et al., 1996*). Hypermethylation in PTPRN was associated with longer progression-free survival in ovarian cancer (*Bauerschlag et al., 2011*). If that is the case, hypomethylation (upregulated mRNA expression level) in PTPRN may be associated with poor prognosis, which is consistent with our results. NCKIPSD is a protein containing SH3 and proline-rich domains. Reports have shown that NCKIPSD is involved in the maintenance of sarcomeres and assembly of myofibrils into sarcomeres (*Lim et al., 2001*). A very recent study reported that NCKIPSD downregulation and increased $\alpha$-tubulin acetylation promotes stiffness of tumor stroma, which in turn, may inhibit chemotherapeutic drug uptake and regulate tumor sensitivity to chemotherapy, resulting in high risk of breast cancer recurrence within 5 years (*You et al., 2017*). Consistently, our findings also showed decreased NCKIPSD expression is associated with high risk of colon cancer recurrence. PGLYRP1 is a member of peptidoglycan recognition proteins which are conserved innate immunity proteins, recognize bacterial peptidoglycan, and play a role in antibacterial immunity and inflammation (*Dziarski & Gupta, 2010*). PGLYRP1 interacts with Hsp70 to induces cytotoxic activity in tumor cells via TNFR1 receptor (*Yashin et al., 2015*). C5orf53 is also called a IgA inducing protein, which enhances IgA secretion from B-cells stimulated via CD40 (*Endsley et al., 2009*). TREML3 is a inhibitory receptor involved in antigen processing (*Cella et al., 1997*). Numerous studies have shown that cancer patients' prognosis and sensitivity to therapy are closely associated with infiltration and density of immunologic cells within primary tumors (*Wels et al., 2008*; *McConnell & Yang, 2009*; *McLean et al., 2011*; *Sethi & Kang, 2011*; *Smith & Kang, 2013*). Of particular note, by applying a novel machine-learning method, called Cell-type Identification By Estimating Relative Subsets of known RNA Transcripts (CIBERSORT), Gentles et al. developed several gene expression signatures to inferring distinct leukocyte subsets representation in bulk tumor transcriptomes (*Gentles et al., 2015*). In several solid tumors including colon cancer, the signatures relating to plasma cells and polymorphonuclear cells were the most significant favorable and adverse module to cancer outcomes, respectively. The broad spectrum involvement of lymphocyte infiltration and intra-tumor immune-suppression implies that this could be the main reason why our 12-gene signature could also predict patient prognosis in several other cancer types including kidney cancer, lung cancer, uveal and skin melanoma, brain cancer, and pancreatic cancer.

Other five genes (NOG, VIP, RIMKLB, NKAIN4, and FAM171B) in the 12-gene signature are functionally sporadic. NOG is related to mesodermal commitment and differentiation pathway (*Costamagna et al., 2016*). High expressing of gene signature including NOG showed a strong trend for a worse prognosis of patients with lung adenocarcinomas (*Rajski, Saaf & Buess, 2015*). VIP, a member of glucagon/secretin superfamily, is the ligand of class II G protein-coupled receptor (*Umetsu et al., 2011*). It causes vasodilation and lowers arterial blood pressure. VIP signaling is enhanced in human prostate cancer (*Fernandez-Martinez et al., 2012*). Elevated VIP secretion is associated with advanced tumor stage in colorectal carcinoma (*Hirayasu et al., 2002*). RIMKLB is involved in alanine, aspartate and glutamate metabolism. RIMKLB up-regulation is associated with radio-resistance in nasopharyngeal carcinomas (*Li et al., 2016*). NKAIN4 may interact

with the beta subunit of Na, K-ATPase (*Gorokhova et al., 2007*). FAM171B which is a single-pass type I membrane protein, belongs to the FAM171 family. It is up-regulated in gemcitabine-resistant pancreatic cancer cell line (*Zhou et al., 2015*). The associations of these genes with cancer and cancer outcomes are very relevant to our findings in this study.

Our signature generated a novel scoring system with relative gene expression values by dividing the raw expression with geometric mean of RPKM values of three house-keeping genes (TFRC, GUSB, and RPLP0). In order to preserve the heterogeneities among tumors to the most extent, ACTB and GAPDH were avoided using as reference genes due to the fact that cytoskeleton and energy metabolism might be greatly deregulated among cancer individuals (*Xiang, Chen & Fu, 2017*; *Stine & Dang, 2013*). A recent study overcomes hypoxia-induced tumor cell resistance by synergistic GAPDH-siRNA and chemotherapy (*Guan et al., 2017*), indicating the important roles of GAPDH in tumor cell resistance. Our normalization process also makes the gene expression scoring system very friendly to different gene expression detection systems including qPCR, RNA-seq, and QuantiGene Plex.

## CONCLUSION

A robust and accurate gene expression signature is essential to assist oncologists to determine which subset of patients at similar TNM stage has high recurrence risk and could benefit from adjuvant therapies. Here we report a new 12-gene expression signature that could separate resectable COAD patients into poor- and good-prognosis group in several independent TCGA and microarray datasets. Functional classification showed that seven of the twelve genes are involved in immune system function and regulation. Our gene expression signature could potentially serve as an effective genomic test in helping identify resectable COAD patients with high risk of poor prognosis. The accuracy and robustness of the signature as a prognostic classification requires further confirmation with large prospective patient cohorts.

### Funding

This work was supported by the National Natural Science Foundation of China (No. 81672334) and the International Science and Technology Cooperation Project of Shanghai (No. 15410710100). The funders had no role in study design, data collection and analysis, decision to publish, or preparation of the manuscript.

### Grant Disclosures

The following grant information was disclosed by the authors:
National Natural Science Foundation of China: 81672334.
International Science and Technology Cooperation Project of Shanghai: 15410710100.

### Competing Interests

The authors declare there are no competing interests.

## Author Contributions

- Dalong Sun, Jing Chen, Longzi Liu, Guangxi Zhao, Pingping Dong and Bingrui Wu conceived and designed the experiments, performed the experiments, analyzed the data, contributed reagents/materials/analysis tools, prepared figures and/or tables, authored or reviewed drafts of the paper, approved the final draft.
- Jun Wang conceived and designed the experiments, contributed reagents/materials/-analysis tools, authored or reviewed drafts of the paper, approved the final draft.
- Ling Dong conceived and designed the experiments, authored or reviewed drafts of the paper, approved the final draft.

## Microarray Data Deposition

The following information was supplied regarding the deposition of microarray data:

RNA-seq data and clinic information for all cancer types were obtained from the Cancer Genome Altas (TCGA) RNA-seq database (https://cancergenome.nih.gov/). Microarray expression data GSE39582, GSE17538 and clinic information for COAD patients were retrieved from Gene Expression Omnibus (GEO) database (https://www.ncbi.nlm.nih.gov/geo/).

## Data Availability

Detailed clinical information of TCGA patients enrolled in phase I and phase II training stage were listed in Tables S1 and S2, respectively.

The detailed clinical information of TCGA patients enrolled in phase I and phase II training stages are listed in Tables S1–S3.

## Supplemental Information

Supplemental information for this article can be found online at http://dx.doi.org/10.7717/peerj.4942#supplemental-information.

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
