# Peer review of "Establishment of a 12-gene expression signature to predict colon cancer prognosis"

_PeerJ, doi:10.7717/peerj.4942_

## Round 0.1 · original submission · Major Revisions

I have now received comments from two expert reviewers who generally support publication of your manuscript but have raised issues that you will need to address.

Please pay very careful attention to each point the reviewers have raised. They have spent considerable time and effort in reviewing your manuscript and you should see their comments as being helpful for you to convey your message appropriately and accurately.

Reviewer 1 ·

Basic reporting

The manuscript requires some editing of English language, grammar and phrasing to improve clarity of meaning. For example correct application of singular and plural rules.
Please explain abbreviations at first use in manuscript e.g. TNM, TCGA in abstract. Some explanation of the GSE datasets should be added to the abstract to clarify.
Appropriate introduction and background provided to the topic, but note comment above on English language use.
Figures are relevant, but issues of resolution should be addressed in the high density figures e.g. Figure 2. Text resolution in Figure 3 plot is poor. Further description and explanation of terms should be added to figure legends e.g. abbreviations used, axis labels. Raw data files are provided. The statistical plots and test results displayed in the figures need further explanation for those not familiar with their use.

Experimental design

The use of available data from TCGA and GEO is described in detail. There appears to be rigorous validation using patient data. However, full assessment of the statistical approaches applied is beyond my area of expertise.

Validity of the findings

The impact and novelty of the 12 gene signature is clearly outlined and potential application and benefits explained in colon and other cancers. Identification of potential genes implicated in progression and metastasis provide useful preliminary information for focus of further study of the factors influencing prognosis, cancer progression and metastasis.

·

Basic reporting

No Comment

Experimental design

No Comment

Validity of the findings

No Comment

Additional comments

In this manuscript “Establishment of a 12-gene expression signature to predict colon cancer prognosis” from Dalong Sun et al., the authors outline a bioinformatics approach to identify poor prognostic group from a general colorectal cancer population. This is an area of importance currently, given the availability of transcriptional and molecular data, in this regard the author add to the already well-established literature in the development of prognostic biomarkers for colorectal cancer. The author have used multiple data sets for discovery and validation, in addition to an extended collection of non-CRC cohorts to demonstrate the prognostic power of their classifier.
The manuscript presents a clear methodology that combines a number of well established approaches to prognostic classification, I do however have a number of points concerning the clinical and pathological status of the patient cohorts selected, which could potentially undermine the approach taken by the authors.

Phase I and Phase II patient cohorts:
Can the authors outline details in a table to clarify the stage of the patients in the training prognostic groups; is it that the poor prognosis group are all late stage, with the good prognosis group being early stage?

As it stands, the classifier is potentially a signature of tumour stage, which is an already well established prognostic indicator as detailed by the authors in their introduction. Again further detailed information needs to be explained here in both Phase I and Phase II.

Can the authors define in a table if these patient received treated at any stage of their disease, and if so was this adjuvant, neo-adjuvant or in the metastatic setting, or have they received surgery alone; factors which have strong bearing on disease outcome. The patient sets used at this stage needs to be detailed in a clear table, a link to the Cancer Genome atlas of GEO is insufficient.

Patient cohort selected.
The authors state in the first 2 lines of their abstract that - “Robust and accurate gene expression signature is essential to assist oncologists to determine which subset of patients at similar TNM stage has high recurrence risk” – also on line 54-56 in the introduction the authors again highlight the importance of this “However, due to the high heterogeneity of disease, the patients at similar stage behave differently in terms of recurrence and response to chemotherapy often differs.” – based on these key points, the author correctly describe that the generation of a signature that can distinguish prognosis within a defined TNM stage is therefore essential (stage II only or stage III only or stage II/III), and not just identify stage which is already defined by TNM.

In order to align this study with the key points raised by the authors within their manuscript, can the authors focus on a single stage, or indeed combine stage II/III and carry out their prognostic analysis; ie does this signature distinguish poor prognostic patients from within a similar stage?

In Figure 4 the authors clearly show that the signature is associated with worst outcome. As discussed above, if this was developed using a defined TNM stage and then tested in an independent cohort of the same specific TNM stage, it would strengthen the validity of the findings. Line 37-39 detail that “Interestingly, advanced stage patients were significantly enriched in high 12-gene score group (Fisher’s exact test p=0.0003).” indicates this may be the case.


The signature generated:
In the first 5 lines of the abstract the authors state: “Here we applied two steps of supervised machine-learning method and established a 12-gene expression signature to precisely predict colon adenocarcinoma (COAD) prognosis by exhaustively using expression of all genes and associated clinic information of TCGA COAD patients” – it is rather misleading to suggest the clinical factors have been included in the machine learning approach to develop the signature – please clarify this sentence or else combine in the clinical information into the machine learning approach.

From figure 1 (which is the extended signature) it appears that it is increased gene expression that is associated with high-risk – is this true for the 12 genes? Is the signature upregulated or down regulated when identifying good or poor prognostic patients?


Functional classification of the signature:
The author’s state in their conclusion, line 278, that “Functional classification” of the genes was carried out, which is not detailed in this study, please include these results in detail.

The 12 gene signature is not listed in the results, apart from being discussed; as this is the key piece to this study more focus needs to be put on the genes themselves – have they been previously associated with prognosis, what is their underlying biology.. etc. Functional classification will require potentially pathway analysis or other methods to characterise the biology.

In addition, if this is an “immune” signature, how does this overlap with the CMS1 subtype identified by Nat Med. 2015 Nov;21(11):1350-6 or other well established immune signature in CRC?

Validation cohorts:
The study is strengthened by the addition of multiple validation cohorts, although similar points raised in the training cohort are present in these cohorts.
For GSE39582; the 566 patients in this cohort again span all stages with mixture or treated and untreated patients – to address the points above the use of a specific stage with a similar treatment schedule (perhaps untreated only) would address concerns.

Furthermore, surprisingly the authors use only the pMMR sub-set of this cohort; if showing this analysis it is also required to show the dMMR (and combined) data also for comparison. Similar comments for GSE17538. In both cases, patient set used at this stage needs to be detailed in a clear table with clear indication what the characteristic are of the entire cohort alongside the identified high- and low-risk group.

Other non-CRC cohorts:
The authors should be commended for applying their signature to other disease types, where they find prognostic value outside of COAD – given the presence of 7 immune genes in their signature, these results are reflective of he importance iof the immue system in the prognosis of cancer – discussion of these points to include Nature Medicine volume 21, pages 938–945 (2015) should be included here.

Introduction lines 50-53 describe classical genetic markers of CRC; as this study is focussed on transcriptional approaches to subtyping cancer, the authors will need to include the CMS Nat Med. 2015 Nov;21(11):1350-6. transcriptional classification system in the introduction and discussion.

---

## Round 0.2 · Minor Revisions

I agree with the reviewer that a short discussion of the functions of the genes identified, in terms of why they represent prognostic markers, would be a useful addition to the manuscript and help readers to understand the value of your work.

I encourage you to add a few short sentences, with appropriate citations.

·

Basic reporting

no comment

Experimental design

no comment

Validity of the findings

no comment

Additional comments

The authors have revised the manuscript and have satisfactory addressed the major points raised during the initial review. If possible (within the word-count limitations) if a small concise description of the specific genes/biology could be included in the discussion this would be welcome.

---

## Round 0.3 · accepted · Accept

Many thanks for submitting your revision and including a few sentences regarding the potential functional roles of the identified genes.